# Determinants of multidrug-resistant urinary tract infections: A retrospective cross-sectional study from a tertiary care hospital in southern Bangladesh

Ibrahim Khalil[1], Abu Sayed[2*], A. K. M. Akbar Kabir[3*], Md. Nurul Alam[1], S. M. Iqbal Hossain[4], Rahima Akther Dipa[5], Md Tanvir Rahman[6]

**1** Field Disease Investigation Laboratory, Department of Livestock Services, Barishal, Bangladesh, **2** Faculty of Animal Science and Veterinary Medicine, Patuakhali Science & Technology University, Barishal Campus, Babugonj, Barishal, Bangladesh, **3** Department of Microbiology, Sher-E-Bangla Medical College, Barishal, Bangladesh, **4** Department of Physiology and Pharmacology, Patuakhali Science & Technology University, Barishal Campus, Babugonj, Barishal, Bangladesh, **5** Department of Physiology, Pharmacology & Toxicology, Habiganj Agricultural University, Habiganj, Bangladesh, **6** Department of Microbiology and Hygiene, Faculty of Veterinary Science, Bangladesh Agriculture University, Mymensingh, Bangladesh

* a.sayedpstu@gmail.com (AS), ak_sb31st@yahoo.com (AKMA)

## Abstract

Multidrug resistance (MDR) in urinary tract infections (UTIs) presents a growing global health threat, particularly in resource-limited settings like Bangladesh, where context-specific data remain limited. This retrospective cross-sectional study, conducted from January to December 2023 at a tertiary care hospital in Barishal, Bangladesh, aimed to identify key predictors and risk factors associated with MDR in UTI patients. Of 1,670 urine samples received, 229 with significant bacterial growth were included for antimicrobial susceptibility testing using the disk diffusion method. *E. coli* (55.9%) was the most common isolate, followed by *Pseudomonas* spp. (20.5%), *Klebsiella* spp. (14.8%), and *Acinetobacter* spp. (8.7%). Of these isolates, 70 (30.56%) were found to be MDR-positive. Multivariate logistic regression revealed that male patients (aOR = 2.2; $p < 0.05$), samples from specialized units (SUs) (aOR = 6.1; $p < 0.001$), and private medical settings (PMSs) (aOR = 3.1; $p < 0.05$) were independently associated with increased odds of MDR. Notably, male patients from PMSs showed significantly elevated MDR risk (aOR = 21.8; $p < 0.05$), indicating a strong predictor for MDR-UTIs. Compared to *Acinetobacter* spp., *E. coli* (aOR = 0.3; $p < 0.05$) and *Pseudomonas* spp. (aOR = 0.1; $p < 0.001$) demonstrated lower MDR odds. Multiple Correspondence Analysis (MCA) showed that MDR-positive status and SUs were the main factors contributing to variations in the data, whereas *Acinetobacter spp.* and *Pseudomonas spp.* contributed to secondary patterns of variation. These findings underscore the multifactorial nature of MDR-UTIs and emphasize the critical role of healthcare-associated exposure and bacterial species

**Data availability statement:** The data supporting the findings of this study are available at https://doi.org/10.5281/zenodo.17605137.

**Funding:** The author(s) received no specific funding for this work.

**Competing interests:** The authors have declared that no competing interests exist.

in MDR development. Targeted antimicrobial stewardship, enhanced surveillance, and evidence-based interventions are urgently needed to curb MDR in UTIs within Bangladesh and similar low-resource contexts.

## Introduction

Multidrug resistance (MDR), a type of antimicrobial resistance (AMR), affects human health, the environment, and the economy. AMR happens when microorganisms become resistant to different types of antimicrobial drugs, such as antibiotics, antivirals, antifungals, and antiparasitics. MDR refers to microorganisms that can resist at least one drug from three or more different groups of antimicrobials [1]. MDR increases both morbidity and mortality by making standard antibiotic treatments ineffective, which leads to prolonged illnesses and higher rates of treatment failure [2]. A study from the World Bank reported that AMR could lower GDP by 2.45% in low-income nations and lead to an alarming global economic shortfall of $16.7 trillion by 2050 (World Bank, 2016) [3]. It also harms ecosystems through contamination via wastewater, agricultural runoff, and improper disposal methods, facilitating horizontal gene transfer that further spreads resistance [4]. In healthcare settings, inanimate surfaces like bedrails, stethoscopes, and supply carts are often contaminated through patient shedding or contact with healthcare workers, facilitating the spread of multidrug-resistant organisms and reducing treatment effectiveness [5,6]. Therefore, the World Health Organization (WHO) has identified AMR as one of the top ten global public health threats. It urgently calls for joint international efforts to reduce its growing impact on health, the environment, and the economy [7].

Urinary tract infections (UTIs) are among the most widespread bacterial infections globally, impacting around 150 million people each year in community settings [8]. They are also the fifth most common type of healthcare-associated infection (HAI) and have a major impact on global mortality, causing about 62,700 deaths in 2015 [9]. The main causative agent of UTIs is uropathogenic *Escherichia coli* (UPEC), followed by pathogens such as *Klebsiella pneumoniae*, *Proteus mirabilis*, *Enterococcus faecalis*, and various *Staphylococcus* spp. [10]. The prevalence of uropathogens varies depending on age, sex, and other factors, with notably higher infection rates in females and significant association seen in individuals aged 19 years old or older [11,12]. Standard treatment for UTIs typically involves antibiotic classes like trimethoprim-sulfamethoxazole, fluoroquinolones, nitrofurantoin, fosfomycin, and β-lactams, with the success of treatment hinging on achieving effective antimicrobial levels in the urine [13–15]. Nevertheless, the growing occurrence of MDR uropathogens poses a serious challenge to effective treatment. Alarmingly, high rates of MDR have been recorded among *E. coli* (77.6%), *Pseudomonas* spp. (90.5%), *Acinetobacter baumannii* (88.5%), and *Klebsiella* spp. (7.3-100%) [16–18]. The high frequency of MDR pathogens highlights the pressing need for thorough surveillance, improved AMS, and the exploration of alternative treatment options to address this global health issue.

MDR-UTIs result from a complicated interaction of demographic, clinical, and environmental elements, creating a substantial challenge for healthcare systems

worldwide. Age and sex are significant predisposing factors, as individuals over 65 years and males show higher prevalence rates [19,20]. Other associated risk factors for MDR-UTIs include prior antibiotic use, recent hospitalization, urinary catheterization, nursing home residence, diabetes mellitus, and recurrent infections [21–23]. The clinical origin of urine samples can influence MDR trends; Shakya *et al.* (2021) reported a higher frequency of MDR isolates in samples obtained from inpatient departments (e.g., medicine, surgery, nephrology, obstetrics) compared to outpatient clinics, with rates of 66.2% and 49.5%, respectively [24]. Additionally, environmental factors play a role; for instance, warm weather may be associated with the occurrence of MDR, as rising temperatures can affect bacterial behavior and promote the development of antibiotic resistance patterns [25].

Despite the well-documented global impact of MDR, there remains a critical gap in understanding the interplay of demographic, clinical, and environmental factors contributing to resistance, especially in low- and middle-income countries (LMICs). In the south-western part of Bangladesh, no study has been conducted to identify the factors associated with MDR, which is crucial for local AMS programs. Moreover, the MDR pattern is alarmingly changing over time and across locations, hindering targeted initiatives [26]. Therefore, this research was undertaken with the aim to determine the predictors and risk factors associated with MDR in UTIs in southern part of Bangladesh and provide evidence-based insights to inform AMS programs tailored to LMICs. The findings have far-reaching implications, offering a model for similar investigations in resource-constrained settings and contributing to the global fight against antimicrobial resistance.

## Methods

### Ethics statement

Ethical approval for this study was obtained from the Institutional Review Board (IRB) of SBMC, affiliated with the University of Dhaka, under memo number 59.14.0000.130.99.001.24.2763, dated 28/11/2024. The study involved the retrospective analysis of fully anonymized secondary data obtained from laboratory records of urine samples submitted for diagnostic purposes.

As the data were de-identified and no interventions or direct patient interactions occurred, the IRB granted a waiver of informed consent. All procedures adhered to the institutional ethical guidelines and complied with the principles outlined in the Declaration of Helsinki.

### Study location and design

A retrospective cross-sectional study was conducted using an anonymized secondary dataset comprising urine culture and sensitivity test results collected between January and December 2023 at Sher-E-Bangla Medical College & Hospital (SBMCH), Barishal, a government-operated tertiary medical care facility in southwestern Bangladesh Fig 1. From January to December 2023, a total of 1,797 urine samples were initially collected from patients suspected of UTIs as part of routine clinical diagnostics. Among these, 1,420 samples originated from the outpatient department (OPD), 180 from specialized units (SUs), 70 from private medical settings (PMSs), and 127 lacked source specification in the dataset. Incomplete records lacking essential demographic or clinical data were excluded from the analysis. The samples exhibiting significant bacterial growth, were included for further microbiological analysis (n = 229). The present study did not involve direct contact with patients or any additional sample collection beyond routine clinical procedures.

### Sample collection and preparation

At first, midstream, clean-catch urine samples were obtained from each patient using wide-mouth, screw-capped, sterile containers, adhering to standard precautions to prevent contamination. Urine samples, each with a minimum volume of 5 mL, were collected from each patient and transported to the microbiology laboratory within 30 minutes of collection. The samples were inoculated into Sheep Blood Agar and MacConkey Agar media and then incubated aerobically at 37 °C for 16–18 hours [27]. Bacterial growth was considered significant if the colony count was ≥ $10^5$ CFU/mL, indicative of a UTI [28].

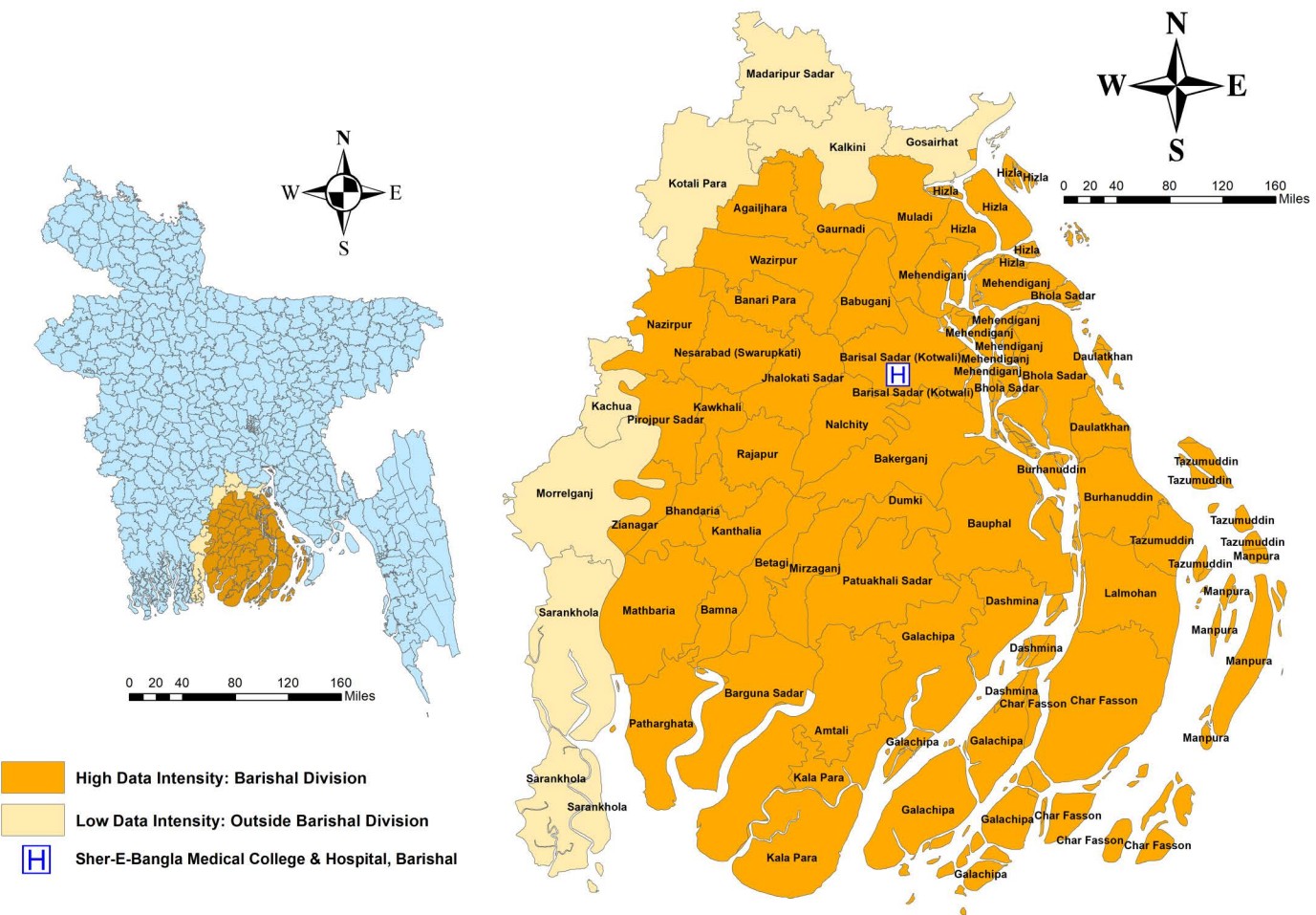

**Fig 1. Geographical Distribution of obtained Urine Samples from UTI Patients across Barishal Division and Surrounding Areas, Bangladesh.**
Base map (administrative boundary) obtained from DIVA-GIS (https://www.diva-gis.org/data.html). Map prepared using ArcGIS version 10.8.

## Laboratory analysis

**Bacterial isolation and identification.** Isolation and identification of bacterial species were based on culture, Gram's staining and biochemical test as described by Cheesbrough, M (1985) [29]. In brief, urine samples with significant bacterial growth were first assessed through Gram staining to classify the isolates as Gram-negative or Gram-positive. For Gram-negative bacteria, *E. coli* was identified based on its ability to ferment lactose, producing pink colonies on MacConkey agar. It was further confirmed through biochemical tests: indole positive, methyl red positive, citrate negative, urease negative, and motile, which helped distinguish it from other Gram-negative bacteria such as *Klebsiella* spp., *Pseudomonas* spp., and *Acinetobacter* spp.. Specifically, *Klebsiella* spp. was differentiated by its non-motility, citrate positive, and indole negative characteristics; *Pseudomonas* spp. by its oxidase positive and non-lactose fermenting properties, along with its production of a characteristic green pigment; and *Acinetobacter* spp. was identified as non-motile, oxidase negative, and urease negative. For Gram-positive bacteria, species were confirmed using catalase and coagulase tests to differentiate *Staphylococcus* spp. from other potential Gram-positive organisms. In our study, no definitive molecular tests were performed. The identification of bacterial isolates was based solely on cultural characteristics and biochemical testing.

**Antimicrobial susceptibility testing (AST).** Bacterial colonies were transferred to tryptic soy broth and incubated at 37°C to achieve a turbidity equivalent to the 0.5 McFarland standard, ensuring standardized inoculum density. A sterile cotton swab was dipped into the prepared suspension, excess fluid was removed, and the swab was used to uniformly streak the surface of Mueller-Hinton agar plates in three directions. Antibiotic discs were placed on the agar surface using a sterile dispenser, and the plates were incubated at 37°C for 16–18 hours. Zones of inhibition were measured and interpreted according to the Clinical Laboratory Standards Institute (CLSI) guidelines, 2022. AST was performed using a broad range of antibiotics, categorized by class: aminoglycosides (amikacin, gentamycin, tobramycin, netilmycin), beta-lactam/beta-lactamase inhibitor combinations (amoxicillin-clavulanate, piperacillin-tazobactam), penicillins (ampicillin, penicillin), cephalosporins (cefazolin, cefepime, cefixime, cefotaxime, cefoxitin, ceftaroline, ceftazidim-avibactam, ceftazidime, ceftriaxone, cefuroxime), carbapenems (imipenem, meropenem), quinolones (nalidixic acid), fluoroquinolones (ciprofloxacin, levofloxacin), macrolides (azithromycin, erythromycin), tetracyclines (doxycycline, tetracycline, tigecycline), glycopeptides (vancomycin), sulfonamides (sulfamethoxazole-trimethoprim), phenicols (chloramphenicol), lincosamides (clindamycin), polypeptides (colistin), fosfomycin, trimethoprim, and oxazolidinones (linezolid). However, due to limitations in the availability of certain antibiotic discs, uniform drug sensitivity testing could not be performed for all the isolates.

## Data cleaning and management

Data from laboratory findings and patient records were entered into an Excel spreadsheet and carefully cleaned to ensure accuracy and completeness. Only urine samples with single bacterial colonies were considered for analysis. Antibiotic names were standardized and categorized into their respective antibiotic classes for determination of MDR. Variables were eventually recoded and then coded for further statistical analysis.

## Data analysis and visualization

**Descriptive analysis.** Descriptive analysis was performed to determine the proportions and 95% confidence intervals (CIs) of demographic, seasonal, and microbial variables, including sex, age, season of sample collection, sample source, and bacterial growth. Proportions were calculated as percentages of the total sample size (N = 229) and summarized in tabular format for clear presentation.

**Univariate analysis.** The study assessed the univariate association between multidrug-resistant (MDR) status and various predictors and risk factors. Chi-square tests were used to evaluate the significance of differences in proportions between MDR-positive and MDR-negative cases for categorical variables. Fisher's exact test was applied where expected cell frequencies were less than 5. The variables analyzed included Season (categorized as Winter, Spring, Summer, and Autumn, where Winter includes December, January, and February; Spring encompasses March, April, and May; Summer covers June, July, and August; and Autumn comprises September, October, and November), Age Group (1–18 years [Child], 19–35 years [Young Adult], 36–55 years [Middle-aged Adult], 56–74 years [Older Adult], and 75 + years [Elderly]), Sex (Female, Male) andSource of Sample (OPD, PMSs, and SUs), with SUs comprising special units within Sher-E-Bangla Medical College such as the Female Surgery Unit, Male Surgery Unit, Pediatric Unit, Labor Unit, Medicine Unit, Female Medicine Unit, Urology Unit, and other departments dedicated to specialized care. These specialized units, which involve hospitalized patients, may carry specific risk factors such as urinary catheterization and extended hospital stays. We considered the bacterial growth of *Acinetobacter* spp*., E. coli, Klebsiella* spp*., and Pseudomonas* spp. for our further analysis. Statistical significance was set at $p < 0.05$ and $p < 0.001$. Results were presented as percentages and counts of MDR-positive and MDR-negative cases, with Pearson Chi-square values used to assess the strength of associations.

**Multivariate analysis.** A multivariate logistic regression model was used to identify independent predictors of MDR-UTIs, adjusting for potential confounders. The model incorporated the variables previously described in the univariate analysis, including age group, sex, bacterial isolates, sample source, and season of sample collection.

The source of sample was included to account for variations in the risk of MDR-UTIs across different patient populations, particularly those from SUs, such as surgery and urology departments, which may be associated with higher risks due to healthcare-related factors such as urinary catheterization and extended hospital stays. While the univariate analysis demonstrated associations between the source of sample and MDR-UTI status, the multivariate approach allowed for control of other variables, thereby providing a clearer understanding of the independent contribution of sample source to MDR-UTI risk.

Adjusted odds ratios (ORs) and 95% confidence intervals (CIs) were calculated for each predictor to assess the strength of association with MDR-UTI. Interaction terms were introduced to examine potential effect modification between key predictors (e.g., the interplay between bacterial species and age group), which could further influence the likelihood of MDR-UTIs. This allowed for a more nuanced understanding of the relationships between variables, beyond simple univariate associations. Model evaluation was possessed using Hosmer-Lemeshow test to assess goodness-of-fit, followed by sensitivity, specificity, positive predictive value (PPV), and negative predictive value (NPV) to evaluate classification performance. Predicted probabilities for MDR-positive classification were generated, and the default classification threshold of 0.5 was adjusted to 0.4 and 0.3 to improve sensitivity. Receiver operating characteristic (ROC) analysis was conducted to visualize model performance, and the area under the ROC curve (AUC) was calculated to measure discriminatory power. Multicollinearity among factors was assessed using variance inflation factors (VIFs). Additionally, the Net Reclassification Index (NRI) and Integrated Discrimination Improvement (IDI) were calculated to evaluate improvements in classification and discrimination after the inclusion of interaction terms.

**Multiple correspondence analysis (MCA).** Multiple Correspondence Analysis (MCA) was employed to examine the associations and underlying structure among categorical variables potentially associated with multidrug-resistant urinary tract infections (MDR-UTIs). MCA is a dimensionality reduction technique specifically designed for categorical data, allowing for the visualization and interpretation of complex relationships between multiple variables.

In this study, each categorical variable was first transformed into a series of binary (dummy) variables. A Burt matrix—a symmetrical matrix summarizing all pairwise cross-tabulations of categories—was then constructed. To extract the principal dimensions, Singular Value Decomposition (SVD) was applied to the Burt matrix. This process decomposes the data into orthogonal dimensions (or components), with each dimension capturing a proportion of the total inertia (variance) in the dataset.

The first dimension (Dimension 1) explains the greatest proportion of the total variance, followed by subsequent dimensions (e.g., Dimension 2) capturing progressively less. The contribution of each category to these dimensions was quantified by evaluating their coordinates along the respective axes, weighted by their inertia. Categories with higher contributions are considered more influential in defining the structure and relationships within the dataset.

Eigenvalues derived from the SVD process were used to determine the proportion of total variance explained by each dimension. Dimensions explaining a cumulative variance of over 60–70% were considered sufficient to capture the major patterns within the data. The resulting graphical and numerical outputs facilitated the identification of clusters and associations between patient characteristics, sample sources, bacterial isolates, and MDR status.

All the statistical analyses were performed using the Statistical Software for Data Science-STATA (version 13, StataCorp, USA).

**Geographic mapping.** The figures for this study were created using Canva, and the study map was developed using Geographic Information System software (ArcGIS, version 10.8, Esri, Redlands, CA, USA).

## Results

### Demographic, seasonal, and isolated bacterial profile

The majority of samples were from females (71.2%), young adults (36.2%), autumn (30.1%), and the OPD (65.5%), compared to males, other age groups, seasons, and sample sources. *E. coli* was the predominant bacterial isolate (55.9%), followed by *Pseudomonas* spp. (20.5%), *Klebsiella* spp. (14.8%), and *Acinetobacter* spp. (8.7%) **Table 1**.

**Table 1. Demographics, Seasonality, and Bacterial Isolates of Urine Samples from UTI Patients in 2023 (N = 229).**

| Variable | Category | % (n) | 95% CI |
|---|---|---|---|
| Sex | Female | 71.2 (163) | 0.65, 0.77 |
| | Male | 28.8 (66) | 0.23, 0.34 |
| Age (Years) | Child (1–18) | 10.9 (25) | 0.06, 0.15 |
| | Young Adult (19–35) | 36.2 (83) | 0.30, 0.42 |
| | Middle-aged Adult (36–55) | 24.0 (55) | 0.18, 0.29 |
| | Older Adult (56–74) | 20.1 (46) | 0.14, 0.25 |
| | Elderly (75+) | 8.7 (20) | 0.05, 0.12 |
| Season | Autumn | 30.1 (69) | 0.24, 0.36 |
| | Spring | 27.1 (62) | 0.21, 0.32 |
| | Summer | 21.8 (50) | 0.16, 0.27 |
| | Winter | 21.0 (48) | 0.15, 0.26 |
| Sample Source | OPD | 65.5 (150) | 0.59, 0.71 |
| | PMSs | 17.9 (41) | 0.13, 0.23 |
| | SUs | 16.6 (38) | 0.12, 0.21 |
| Bacterial Isolates | *E. coli* | 55.9 (128) | 0.49, 0.62 |
| | – OPD | 70.3 (90) | |
| | – PMSs | 14.8 (19) | |
| | – SUs | 14.8 (19) | |
| | *Pseudomonas* spp. | 20.5 (47) | 0.15, 0.25 |
| | – OPD | 51.1 (24) | |
| | – PMSs | 21.3 (10) | |
| | – SUs | 27.6 (13) | |
| | *Klebsiella* spp. | 14.8 (34) | 0.10, 0.19 |
| | – OPD | 61.8 (21) | |
| | – PMSs | 23.5 (8) | |
| | – SUs | 14.7 (5) | |
| | *Acinetobacter* spp. | 8.7 (20) | 0.05, 0.12 |
| | – OPD | 75.0 (15) | |
| | – PMSs | 20.0 (4) | |
| | – SUs | 5.0 (1) | |

## Univariate association

In our study, 30.56% (70) of the samples were identified as MDR-positive **Fig 2**. Significant associations were observed between MDR prevalence and age, sex, and source of sample. The Elderly (75 + years) group had the highest MDR prevalence (50%), followed by the Older Adult (56–74 years) group (45.7%) **Table 2**. Males exhibited a significantly higher MDR rate (50%) compared to females (22.7%). Among the sample sources, specialized units had the highest MDR prevalence (57.9%), followed by private practice (46.3%), while OPD samples showed the lowest MDR rate (19.3%).

## Multivariate relationship

**Table 3** summarizes the multivariate logistic regression analysis identifying factors associated with MDR status. Male patients had significantly higher odds of MDR infection (aOR = 2.2; $p < 0.05$). Samples from PMSs and SUs were significantly associated with increased MDR risk compared to OPD samples (aOR = 3.1 and 6.1, respectively; $p < 0.001$ and $p < 0.05$). Compared to *Acinetobacter* spp., *E. coli* (aOR = 0.3; $p < 0.05$) and *Pseudomonas* spp. (aOR = 0.1; $p < 0.001$)

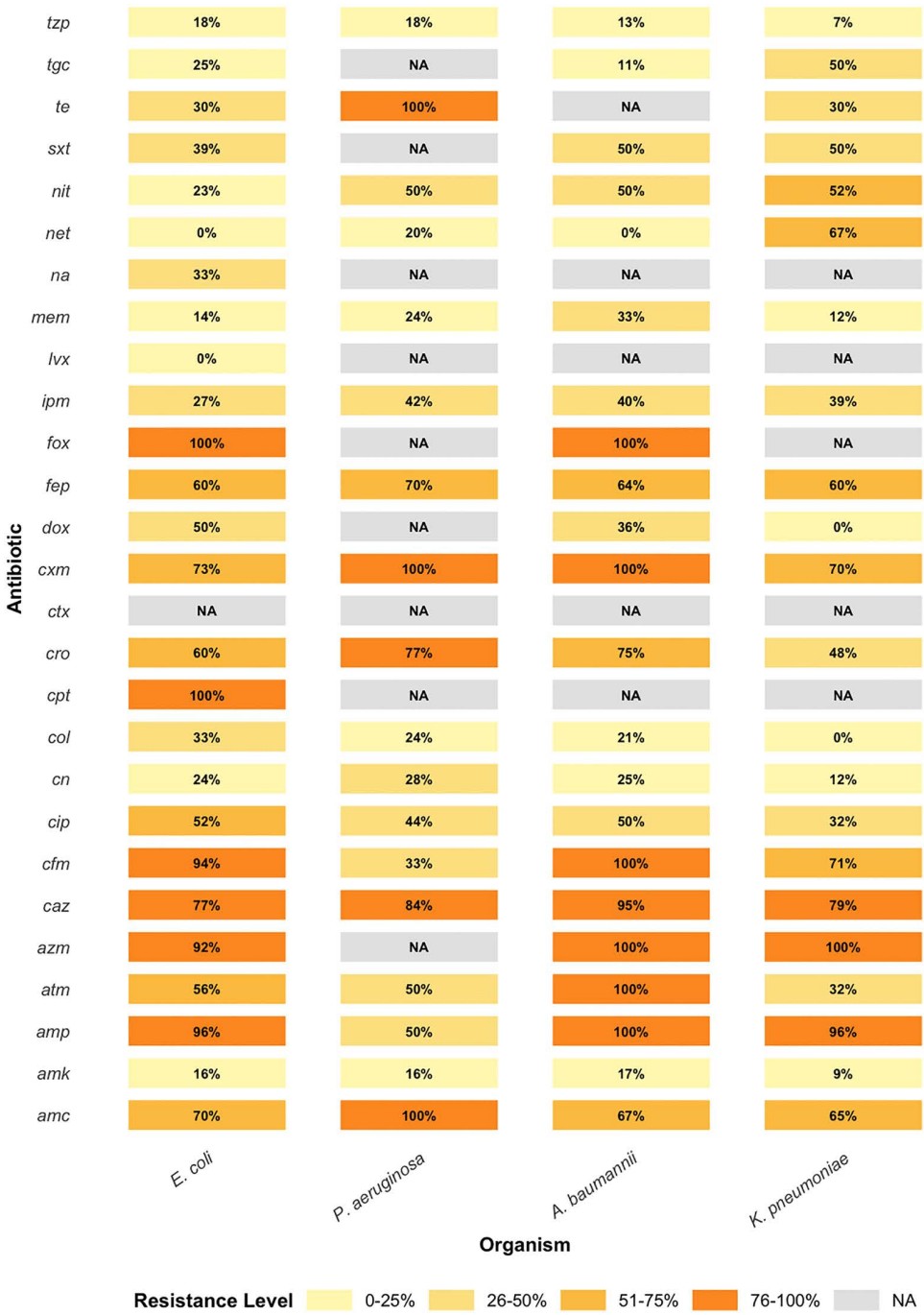

**Fig 2. Antibiotic resistance patterns of bacterial isolates, illustrating the percentage of resistance to each tested antibiotic.** The resistance profiles for each antibiotic are presented as percentage values, showing the extent to which each bacterial strain is resistant to the specific antibiotics tested. The antibiotics tested: amoxicillin-clavulanate (amc), amikacin (amk), ampicillin (amp), aztreonam (atm), azithromycin (azm), ceftazidime (caz), cefixime (cfm), ciprofloxacin (cip), gentamycin (cn), colistin (col), ceftaroline (cpt), ceftriaxone (cro), cefotaxime (ctx), cefuroxime (cxm), doxycycline (dox), cefepime (fep), cefoxitin (fox), imipenem (ipm), levofloxacin (lvx), meropenem (mem), nalidixic acid (na), netilmicin (net), nitrofurantoin (nit), sulfamethoxazole-trimethoprim (sxt), tetracycline (te), tigecycline (tgc), and piperacillin-tazobactam (tzp).

**Table 2. Univariate Analysis of Factors Associated with MDR Prevalence (N = 229).**

| Variable | Category | MDR (+), %(n) | MDR (-), %(n) | p-value | Pearson Chi² |
|---|---|---|---|---|---|
| **Season** | Autumn | 29.0 (20) | 71.0 (49) | 0.337 | 3.3748 |
| | Spring | 24.2 (15) | 75.8 (47) | | |
| | Summer | 40.0 (20) | 60.0 (30) | | |
| | Winter | 31.3 (15) | 68.8 (33) | | |
| **Age group** | Child (1–18 years) | 24.0 (6) | 76.0 (19) | **<0.05*** | 11.98 |
| | Young Adult (19–35 years) | 22.9 (19) | 77.1 (64) | | |
| | Middle-aged Adult (36–55 years) | 25.5 (14) | 74.5 (41) | | |
| | Older Adult (56–74 years) | 45.7 (21) | 54.3 (25) | | |
| | Elderly (75 + years) | 50.0 (10) | 50.0 (10) | | |
| **Sex** | Female | 22.7 (37) | 77.3 (126) | **<0.001**** | 16.4974 |
| | Male | 50.0 (33) | 50.0 (33) | | |
| **Source of sample** | OPD | 19.3 (29) | 80.7 (121) | **<0.001**** | 27.0969 |
| | PMSs | 46.3 (19) | 53.7 (22) | | |
| | SUs | 57.9 (22) | 42.1 (16) | | |
| **Bacterial Growth** | *Acinetobacter* spp. | 50.0 (10) | 50.0 (10) | 0.137 | 5.5217 |
| | *E. coli* | 30.5 (39) | 69.5 (89) | | |
| | *Klebsiella* spp. | 32.4 (11) | 67.6 (23) | | |
| | *Pseudomonas* spp. | 21.3 (10) | 78.7 (37) | | |

* *p* is significant at 0.05 level.

** *p* is significant at 0.001 level.

**Table 3. Multivariate Logistic Regression Analysis (MLRA) of Factors Associated with Multidrug-Resistant (MDR) Status.**

| Variable | Category | aOR | 95% CI | p-value |
|---|---|---|---|---|
| Sex | Female | **Ref** | | |
| | Male | 2.2 | 1.1, 4.6 | **<0.05*** |
| Sample Source | OPD | **Ref** | | |
| | PMSs | 3.1 | 1.3, 7.0 | **<0.05*** |
| | SUs | 6.1 | 2.5, 14.6 | **<0.001**** |
| Bacterial growth | *Acinetobacter* spp. | **Ref** | | |
| | *E. coli* | 0.3 | 0.1, 0.8 | **<0.05*** |
| | *Klebsiella* spp. | 0.3 | 0.1, 1.1 | 0.080 |
| | *Pseudomonas* spp. | 0.1 | <0.1, 0.4 | **<0.001**** |
| Sex-Source | Male-SUs | 2.6 | 0.3, 17.8 | 0.326 |
| | Male-PMSs | 21.8 | 2.7, 175.5 | **<0.05*** |

**p* is significant at 0.05 level.

***p* is significant at 0.001 level.

had significantly lower odds of being MDR, while *Klebsiella* spp. showed no statistically significant difference (aOR = 0.3; $p = 0.080$). The interaction term Male-PMSs indicated a notably higher MDR association (aOR = 21.8; $p < 0.05$).

The Hosmer-Lemeshow test demonstrated a good model fit ($p = 0.96$), indicating no significant evidence of poor calibration. At the default threshold of 0.5, the model achieved high specificity (91.8%) while maintaining moderate sensitivity (41.4%), effectively identifying MDR-negative cases without compromising performance. Adjusting the classification threshold to 0.4 increased sensitivity to 55.7%, with a slight reduction in specificity to 85.5%, while further lowering the

threshold to 0.3 increased sensitivity to 70.0%, although specificity decreased to 78.0%, reflecting the trade-off between early detection and false positives. The model's positive predictive value (PPV) was 69.1%, ensuring that 69.1% of predicted MDR-positive cases were true positives, while the negative predictive value (NPV) of 78.1% ensured accuracy in identifying MDR-negative cases. The area under the receiver operating characteristic (ROC) curve (AUC) was 0.77, indicating good discriminatory power, typically considered in the range of 0.7 to 0.8. The inclusion of the interaction term between Sex and Source significantly improved model performance, as confirmed by a likelihood ratio test ($\chi^2 = 8.55$, $p = 0.014$). The Net Reclassification Improvement (NRI) of 0.27 ($p = 0.010$) and the Integrated Discrimination Improvement (IDI) of 0.058 ($p = 0.004$) further suggest significant improvements in risk classification and model discrimination compared to baseline. Multicollinearity was assessed using variance inflation factors (VIFs), with all values below 3.3, which is well below the threshold typically considered problematic (VIF > 5 or 10), suggesting no multicollinearity concerns. These findings highlight the model's potential for clinical application, with tailored threshold selection to balance sensitivity and specificity while maintaining good discrimination and reliability.

### Multiple correspondence analysis (MCA)

To explore the complex interrelationships among categorical predictors associated with multidrug-resistant urinary tract infections (MDR-UTIs), a MCA was performed. The analysis extracted two primary dimensions that jointly explained 78.65% of the total variability in the dataset—Dimension 1 accounted for 75.81%, while Dimension 2 explained an additional 2.84%.

Dimension 1 captured the principal structure of the data and was primarily shaped by categories such as MDR-positive status (Coordinate: -1.464) and samples collected from SUs (Coordinate: -1.950), both of which exhibited strong contributions to this axis. This indicates a strong association between MDR status and certain clinical sources of infection. Conversely, MDR-negative cases (Coordinate: 0.644) and samples from OPD (Coordinate: 0.898) were positioned on the opposite side of this dimension, reflecting contrasting patterns of distribution.

Dimension 2, although accounting for a smaller proportion of variability, highlighted the contribution of specific bacterial isolates. Notably, *Acinetobacter* spp. (Coordinate: 3.928) and *Pseudomonas* spp. (Coordinate: -2.426) exhibited substantial influence on this secondary axis, suggesting their distinct roles in the development of MDR-UTIs.

The MCA plot (**Fig 3**, **Table 4**) provided a visual representation of the proximities and relationships among the different categories. Variables contributing more prominently to the dimensions are considered more influential in determining the overall data structure. These findings underscore critical associations between sample sources, bacterial species, and MDR status, which could inform more targeted surveillance and intervention strategies to combat antimicrobial resistance in clinical UTI cases.

### Discussion

Antimicrobial stewardship (AMS) poses significant challenges particularly in the developing countries due to complexities in governing the health sector [30]. Inadequate health facilities, lack of strict regulations, and poor awareness hinder effective antimicrobial stewardship, thereby exacerbating the prevalence of MDR organisms [31]. Therefore, our study concluded the critical insights for comprehending the significant predictors and risk factors for MDR patterns in UTI patients from a tertiary care hospital.

In our study, females (71.2%) were more frequently affected by UTIs, which aligns with previous findings [32]. For instance, a study conducted in Aveiro, Portugal, reported that nearly 80% of UTI cases occurred in females, compared to 20.4% in males [33]. Despite this higher incidence, we observed a significantly greater prevalence of MDR-positive cases among males (50%) than females (22.7%) ($p < 0.001$). This finding is further supported by MLRA, where male sex was independently associated with increased odds of MDR infection (aOR = 2.2; 95% CI: 1.1–4.6; $p < 0.05$). This observation may be attributed to the clinical complexity often associated with male UTIs. In contrast to the typically uncomplicated

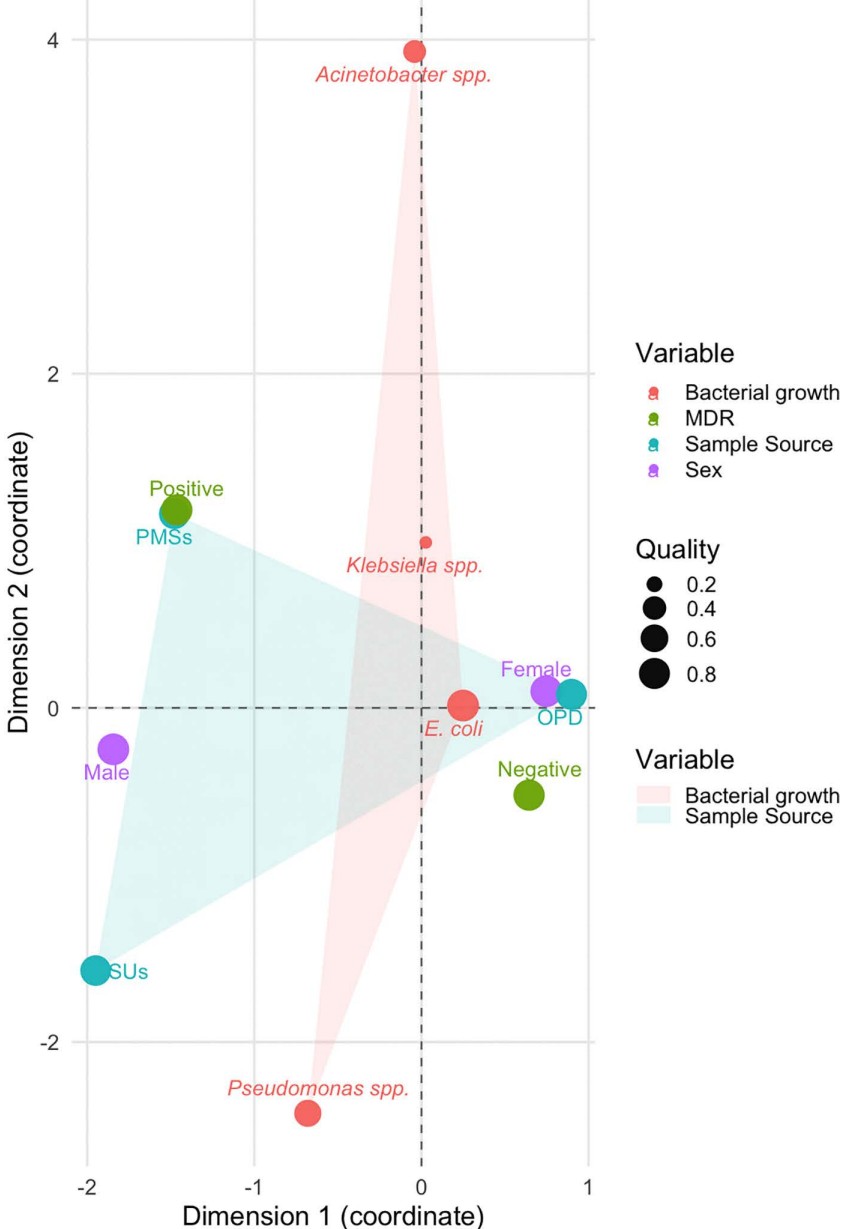

**Fig 3. Multiple Correspondence Analysis (MCA) of Predictors Associated with MDR UTIs.** The plot visualizes the associations between bacterial growth, MDR status, sample source, and patient sex. The first two dimensions explain a cumulative 78.65% of the total variability. The proximity between categories indicates a stronger association, and the size of the points represents the Quality (accuracy) of their representation in this 2D space.

infections seen in females, UTIs in males are frequently linked to underlying urogenital abnormalities, urinary retention, catheter use, or prostatic involvement [34]. These conditions often necessitate prolonged hospitalization and empirical use of broad-spectrum antibiotics, both of which are well-established drivers of MDR [35]. Moreover, male patients are more likely to be admitted to critical care or surgical units—settings characterized by high antibiotic exposure and increased prevalence of resistant pathogens [36].

**Table 4. Contribution and Coordinates of Categories in MCA for MDR status in UTI patients.**

| Variable | Category | Mass | Quality | % Inertia | Dimension 1 (Primary) Coordinate | Contribution | Dimension 2 (Secondary) Coordinate | Contribution |
|---|---|---|---|---|---|---|---|---|
| **Sex** | Female | 0.178 | 0.848 | 0.089 | 0.747 | 0.099 | 0.100 | 0.002 |
| | Male | 0.072 | 0.848 | 0.219 | **-1.844** | **0.245** | -0.248 | **0.004** |
| **Sample Source** | OPD | 0.164 | 0.801 | 0.125 | 0.898 | 0.132 | 0.081 | 0.001 |
| | PMSs | 0.045 | 0.778 | 0.098 | -1.479 | 0.098 | **1.162** | **0.060** |
| | SUs | 0.041 | 0.776 | **0.158** | **-1.950** | **0.158** | **-1.572** | **0.103** |
| **Bacterial growth** | *Acinetobacter* spp. | 0.022 | 0.371 | 0.026 | -0.041 | 0.000 | **3.928** | **0.337** |
| | *E. coli* | 0.140 | 0.854 | 0.008 | 0.249 | 0.009 | 0.014 | 0.000 |
| | *Klebsiella* spp. | 0.037 | 0.182 | 0.006 | 0.026 | 0.000 | 0.990 | 0.036 |
| | *Pseudomonas* spp. | 0.051 | 0.564 | 0.047 | **-0.680** | **0.024** | **-2.426** | **0.302** |
| **MDR** | Negative | 0.174 | 0.813 | 0.069 | **0.644** | 0.072 | **-0.522** | **0.047** |
| | Positive | 0.076 | 0.813 | **0.156** | **-1.464** | **0.164** | **1.185** | **0.107** |

"Mass" indicates the relative weight of each category in the dataset. "Quality" reflects how well the category is represented by the two dimensions. "Co-ord" (Coordinates) represents the position of each category along the primary (Dim 1) and secondary (Dim 2) axes; categories with coordinates far from zero are significant. "Contribution" indicates the relative importance of the category in defining the dimension. Dimension 1 accounts for 75.8% of the total inertia, and Dimension 2 accounts for 2.8%. Subsequent dimensions (including Dimension 3, 0.3%) were excluded as they contributed negligible variance.

In our study, male patients from SUs did not show a statistically significant association (aOR = 2.6; 95% CI: 0.3–17.8; $p = 0.326$), however, male patients from PMSs exhibited significantly higher odds of MDR infection (aOR = 21.8; 95% CI: 2.7–175.5; $p < 0.05$). This finding highlights the context-dependent nature of MDR risk, where healthcare environment, clinical practices, and possible antibiotic stewardship differences in private settings may contribute to elevated resistance.

These results suggest that MDR development is influenced by a combination of factors beyond patient demographics, such as healthcare exposure intensity, antibiotic use patterns, and institutional infection control practices. Therefore, relying solely on sex or other demographic variables may overlook critical determinants of resistance. This multifactorial understanding of MDR is consistent with the perspective presented by Keenan *et al.,* 2024 [37], which emphasizes the need to evaluate both host-related and healthcare-associated risk factors when addressing AMR.

Young adults (36.2%) aged from 19-35 years showed the highest positive cases for UTIs, whereas the infection rate gradually declined in accordance with the age. Interestingly, our study revealed less susceptibility of children to UTIs, aligning with a study carried out in India [38]. This lower prevalence in children can be explained by several factors such as children's immune systems are still in developmental phase, may result in a lower incidence of UTI symptoms or infections during early childhood. The immune response in children may not always trigger noticeable symptoms, making UTIs less prevalent in this age group [39]. In addition, urinary reflux, particularly vesicoureteral reflux (VUR), which is less common in children and is a known risk factor for UTIs. In cases where VUR is absent, the chances of bacterial infection leading to UTIs are reduced, contributing to the lower prevalence observed in children [40]. Furthermore, nutritional status in children is another critical factor influencing UTI prevalence. While this study did not directly assess malnutrition, it remains a significant factor to consider in understanding UTI prevalence in children [41].

On the contrary, a number of studies acknowledged the older age as a significant risk factor for UTIs that are not supporting our current results [42,43]. A statistically significant association was observed between age and MDR status in the univariate analysis ($p = 0.018$), with the highest MDR rates reported among elderly (50.0%) and older adult (45.7%) patients. This trend may reflect the cumulative effects of immunosenescence, increased comorbidities, frequent antibiotic exposure, and prolonged hospital stays — all of which contribute to the selection of MDR pathogens [44]. Despite this, age did not establish as an independent predictor in the multivariate logistic regression model, indicating that its influence

may be mediated through other clinical or healthcare-related variables. In contrast, Milovanovic *et al.* identified age as an independent risk factor for MDR-UTIs, underscoring the context-specific nature of such associations [45].

Previous studies have highlighted seasonal variation as a contributor to UTI prevalence, though, it is not true always [46,47]. Therefore, we considered seasonality as an important factor in relation to UTI occurrences. We observed the highest frequencies in autumn; nevertheless, infection rates across all seasons were almost similar, demonstrating slight variation. In a study of California, Elser and his colleagues identified a clear monotonic inclination in UTI diagnosis in winter when the temperature falls, while the least cases were recorded during summer months [48]. This could be due to increased dehydration and reduced fluid intake in the warmer seasons, resulting in reduction of urine output and subsequently the elimination of uropathogens, thereby acting as environmental risk factors for UTIs [49]. Although, seasonal variation was considered as predictor for UTIs, we could not establish any direct impact of climate on development of MDR pathogens, and therefore, could not determine seasonal variation as an independent risk factor.

The OPD accounted for the highest proportion of UTI-positive samples (65.5%), suggesting that most infections in the present study were community-acquired. However, the likelihood of MDR was significantly higher in samples obtained from SUs and PMSs compared to OPD. The MLRA confirmed that samples from PMSs and SUs had significantly increased odds of MDR (aOR = 3.1, $p < 0.05$; aOR = 6.1, $p < 0.001$, respectively), underscoring the strong influence of healthcare-associated exposure. Notably, an interaction effect between male sex and PMSs was observed, with male patients from PMSs having markedly higher odds of MDR infection (aOR = 21.8; $p < 0.05$), suggesting that institutional practices, empirical antibiotic usage, and complex clinical histories may collectively contribute to elevated MDR risk in this subgroup. This indicates that the healthcare associated factors are a strong independent predictor for MDR-UTIs in the human patients, supported by a number of studies [50–53].

Among the four (4) bacteria identified from the samples, *E. coli* was the most prevalent (55.9%) among the other gram-negative organisms. A comparable study conducted in Iraq identified *E. coli* as the most frequent pathogen (68.3%) which is aligned with our current study finding [52]. Besides that, several other studies have recognized *E. coli* as the paramount contributor for UTIs in human patients which are evident from different healthcare settings throughout the world [54–57].

Our research reveals significant differences in MDR patterns of the bacterial isolates from different clinical settings causing UTIs. *E. coli* (aOR = 0.3; $p = 0.022$) and *Pseudomonas* spp. (aOR = 0.1; $p < 0.001$) exhibited significantly lower odds of MDR than *Acinetobacter* spp., despite the absence of a significant univariate relationship between bacterial species and MDR profile. We used *Acinetobacter* spp. as the baseline in the MLRA model due to its clinical significance in relation to MDR and its role in critical healthcare-associated infections, although *E. coli* was the most dominant organism causing UTIs in our investigation. These results emphasize the importance of implementing effective control strategies against *Acinetobacter* spp. as a notable and prevalent MDR pathogen, as highlighted by previous studies [58–60]. Additionally, the lower odds for *E. coli* and *Pseudomonas* spp. suggest diverse resistance mechanisms, underscoring the need for pathogen-specific treatment protocols and continuous monitoring and research into their resistance patterns to prevent the development and spread of resistant genes and organisms through the agent–host–environment interface.

In this study, we conducted MCA to provide a detailed exploration of MDR patterns revealing that Dimension 1 accounts for 75.8% of the total variance, while Dimension 2 contributed an additional 2.8%. Dimension 1 primarily demonstrates the variability driven by healthcare-associated factors, such as the strong association between MDR-positive status and SUs. This finding underscores the critical role of hospital environments in the emergence and propagation of MDR pathogens. In contrast, Dimension 2 highlights the secondary but significant contributions of specific bacterial isolates, with *Acinetobacter* spp. (33.7%) and *Pseudomonas* spp. (30.2%), emerging as dominant contributors. These results are aligned with global reports identifying these pathogens as high-priority targets for infection control [61–65].

## Strengths and limitations

This study is the first study reported from south-western region of Bangladesh, which employs a robust multifaceted approach to identify independent predictors as well as risk factors for MDR patterns, ensuring deeper insights into healthcare-associated risks and bacterial resistance profiles. It emphasizes pathogen-specific insights, such as resistance patterns in *E. coli*, *Pseudomonas* spp., and *Acinetobacter* spp., aligning with global infection control priorities. Additionally, the incorporation of seasonal variations and alignment with global data enhances the study's relevance and scope for practical applications, particularly in AMS and targeted infection control strategies.

Although the study was conducted in a tertiary care hospital, its findings may not be entirely pertinent to other health-care settings, such as rural or primary care facilities. The limited sample size, restricted geographic representation, and narrow focus on ASTs and detecting uropathogens for MDR patterns constrain its applicability to broader populations. Additionally, certain potential confounders, including patients' nutritional status, co-morbidities, socioeconomic factors, antibiotic regimens and precise antibiotic usage history, were not accounted for due to insufficient data, potentially influencing the interpretation of MDR determinants.Despite assessing seasonality, the study was unable to establish a direct association between seasonal variations and MDR-UTIs, highlighting limitations in integrating environmental data and controlling for confounding climatic factors. Moreover, the cross-sectional design inherently restricts cause-and-effect relationships between the identified predictors and MDR status. Therefore, longitudinal studies are recommended further to establish a strong comprehensive understanding of the association between the predictors and risk factors for causing MDR in UTI patients.

## Recommendations

A comprehensive approach is essential to mitigate the spread and impact of MDR pathogens. Robust AMS programs should enforce stringent antibiotic regulations, enhance public awareness, and promote evidence-based prescribing. Infection control must prioritize high-risk settings through improved hygiene, disinfection, and reduced invasive procedures. Pathogen-specific treatment protocols tailored to resistance patterns, continuous surveillance using advanced molecular tools, and investments in healthcare infrastructure are imperative. Expanding public education on antibiotic use and fostering interdisciplinary research through a One Health approach will further strengthen efforts to combat MDR-UTIs globally.

In conclusion, our study highlights the multifaceted predictors of MDR-UTIs, involving healthcare-associated factors, patient demographics, and bacterial species. While females exhibited a higher prevalence of UTIs, male gender emerged as a significant independent predictor for MDR status. Healthcare environments—specifically Specialized Units (SUs) and Private Medical Settings (PMSs)—were identified as critical drivers of resistance, underscoring the need for targeted interventions in these high-risk settings. Additionally, the significantly higher resistance profile of *Acinetobacter* spp. necessitates pathogen-specific containment strategies. Future longitudinal studies should investigate the specific institutional practices contributing to high MDR rates in SUs and PMSs, to explain the disparity between these settings and the high-volume but lower-risk Outpatient Department.

## Acknowledgments

We sincerely acknowledge the guidance and ethical oversight provided by the Institutional Review Board (IRB) of Sher-E-Bangla Medical College throughout this study. Additionally, we extend our gratitude to the laboratory staff at the Department of Microbiology, Sher-E-Bangla Medical College, Barishal, for their valuable assistance with sample processing and data collection.

During the preparation of this manuscript, the authors utilized ChatGPT (version 4.0) for linguistic improvements. The authors carefully reviewed and revised the content to ensure accuracy, completeness, and adherence to the journal's standards and take full responsibility for the final content of the published article.

## Author contributions

**Conceptualization:** Ibrahim Khalil.

**Formal analysis:** Ibrahim Khalil.

**Investigation:** Ibrahim Khalil, A. K. M. Akbar Kabir.

**Methodology:** Ibrahim Khalil, Abu Sayed, A. K. M. Akbar Kabir.

**Project administration:** A. K. M. Akbar Kabir.

**Resources:** Ibrahim Khalil, Abu Sayed, A. K. M. Akbar Kabir, S. M. Iqbal Hossain.

**Software:** Ibrahim Khalil, Abu Sayed.

**Supervision:** Abu Sayed, A. K. M. Akbar Kabir.

**Visualization:** Ibrahim Khalil, Abu Sayed.

**Writing – original draft:** Ibrahim Khalil, Abu Sayed, A. K. M. Akbar Kabir, Md. Nurul Alam, Rahima Akther Dipa, Md Tanvir Rahman.

**Writing – review & editing:** Abu Sayed.

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
