## [Decision Letter · Decision Letter 0]

24 Sep 2025

PGPH-D-25-02034

Determinants of multidrug-resistant urinary tract infections: a retrospective cross-sectional study from a tertiary care hospital in southern Bangladesh

Dear Dr. Sayed,

Thank you for submitting your manuscript to PLOS Global Public Health. After careful consideration, we feel that it has merit but does not fully meet PLOS Global Public Health’s publication criteria as it currently stands. Therefore, we invite you to submit a revised version of the manuscript that addresses the points raised during the review process.

We look forward to receiving your revised manuscript.

Kind regards,

Delfina Fernandes Hlashwayo, Ph.D.

Academic Editor

Journal Requirements:

1. Please ensure that your Ethics Statement is available in its entirety at the beginning of your Methods section, under a subheading 'Ethics Statement'.

2. Please provide separate figure files in .tif or .eps format.

3. We have noticed that you have two Table 3 in the manuscript file. Please amend the label of the tables and it must be in ascending order.

4. In the online submission form, you indicated that “Data can be made available upon reasonable request to ibrahim.khalil@dls.gov.bd”.

3. Uploaded as supplementary information.

5. Some material included in your submission may be copyrighted. According to PLOS’s copyright policy, authors who use figures or other material (e.g., graphics, clipart, maps) from another author or copyright holder must demonstrate or obtain permission to publish this material under the Creative Commons Attribution 4.0 International (CC BY 4.0) License used by PLOS journals. Please closely review the details of PLOS’s copyright requirements here: PLOS Licenses and Copyright. If you need to request permissions from a copyright holder, you may use PLOS's Copyright Content Permission form.

Potential Copyright Issues:

Figure 1:  please (a) provide a direct link to the base layer of the map (i.e., the country or region border shape) and ensure this is also included in the figure legend; and (b) provide a link to the terms of use / license information for the base layer image or shapefile. We cannot publish proprietary or copyrighted maps (e.g. Google Maps, Mapquest) and the terms of use for your map base layer must be compatible with our CC-BY 4.0 license.

Additional Editor Comments (if provided):

Reviewer #1:

Reviewers' comments:

Reviewer's Responses to Questions

**Comments to the Author**

1. Does this manuscript meet PLOS Global Public Health’s publication criteria?

Reviewer #1: Partly

2. Has the statistical analysis been performed appropriately and rigorously?

Reviewer #1: No

3. Have the authors made all data underlying the findings in their manuscript fully available (please refer to the Data Availability Statement at the start of the manuscript PDF file)?

Reviewer #1: No

4. Is the manuscript presented in an intelligible fashion and written in standard English?

Reviewer #1: No

Reviewer #1: Review Report Manuscript PGPH-D-25-02034

Title: Determinants of multidrug-resistant urinary tract infections: a retrospective cross-sectional study from a tertiary care hospital in southern Bangladesh

General Comments

This manuscript addresses a relevant and timely topic: the determinants of multidrug-resistant (MDR) urinary tract infections in a resource-limited setting. The subject is important for both local and global public health. Data from a tertiary hospital in Bangladesh provide valuable insights.

However, several methodological, analytical, and presentation aspects require clarification or improvement before the article can be considered for publication. Inconsistencies in numbers, insufficient methodological details, and sometimes confusing presentation of results currently limit the understanding and reproducibility of the study.

Major Comments

1. Inconsistencies in numbers

The abstract mentions 1,797 urine samples, whereas the main text reports 1,697. Please check and harmonize these numbers.

2. Study population and inclusion criteria

Inclusion criteria are not clearly described (hospitalized patients? children included? sampling method?). The presence of patients from surgery/urology suggests specific risk factors (surgery, urinary catheters) that should be clarified and discussed.

3. Microbiological methods

- Questionable use of Gram staining on MacConkey agar.

- Gram-positive bacteria: refer to presumptive rather than definitive identification.

- Lack of standardized inocula may bias susceptibility testing.

- Reported incubation time (16–18 h) is shorter than CLSI standards: please verify and reference.

- Antibiotics should be written in lowercase and grouped by class.

- Clearly distinguish quinolones from fluoroquinolones.

4. Definition of multidrug resistance

MDR definition is missing. Please provide the exact definition with an appropriate reference (e.g., Magiorakos et al., 2012, CMI).

5. Presentation of results

- Table 1: Only four species isolated from 229 positive urines seems surprising → clarify.

- Add a table showing susceptibility profiles used to define MDR cases.

- Table 2: presentation unclear → specify whether it compares MDR+ vs MDR− or between categories.

- MCA: mentioned but not shown → add a figure.

6. Interpretation of results

- Lines 341–347: significant association between MDR and male sex is unexpected → provide possible explanations (e.g., catheters, prostatic disease).

- Line 367: low prevalence in children should be explained (immature immunity, reflux, lack of circumcision, malnutrition).

Minor Comments

- Abstract: report MDR prevalence (60/229 = 26.2%).

- Abstract and text: simplify overly long sentences.

- References: provide those for culture media used.

- Abstract: rephrase 'highlighting primary and secondary contributors' for clarity.

- Antibiotics: harmonize formatting (lowercase, grouped by classes).

- Text: shorten some sentences for readability.

Final Recommendation

After carefully reviewing the manuscript, I recognize the importance of the topic, which addresses multidrug-resistant (MDR) urinary tract infections in a resource-limited setting. The data presented are relevant for both local and global public health.

However, the current version of the manuscript presents several major methodological and reporting issues that significantly limit its clarity, reproducibility, and scientific contribution. These include inconsistencies in reported numbers, insufficient description of inclusion criteria and microbiological methods, absence of essential results (e.g., antibiotic susceptibility profiles), and unclear presentation of statistical analyses. The interpretation of findings also requires further development and contextualization.

I recommend major revisions before the manuscript can be considered for publication.

**Do you want your identity to be public for this peer review?** For information about this choice, including consent withdrawal, please see our Privacy Policy

Reviewer #1: **Yes:** Cyrille Bisseye, PhD

---

## [Decision Letter · Decision Letter 1]

2 Dec 2025

Determinants of multidrug-resistant urinary tract infections: a retrospective cross-sectional study from a tertiary care hospital in southern Bangladesh

PGPH-D-25-02034R1

Dear Dr. Sayed,

We are pleased to inform you that your manuscript 'Determinants of multidrug-resistant urinary tract infections: a retrospective cross-sectional study from a tertiary care hospital in southern Bangladesh' has been provisionally accepted for publication in PLOS Global Public Health.

Best regards,

Julia Robinson

Executive Editor

Reviewer Comments (if any, and for reference):

Reviewer's Responses to Questions

**Comments to the Author**

Reviewer #1: All comments have been addressed

publication criteria?

Reviewer #1: Yes

3. Has the statistical analysis been performed appropriately and rigorously?

Reviewer #1: Yes

4. Have the authors made all data underlying the findings in their manuscript fully available (please refer to the Data Availability Statement at the start of the manuscript PDF file)?

Reviewer #1: Yes

5. Is the manuscript presented in an intelligible fashion and written in standard English?

Reviewer #1: Yes

Reviewer #1: Dear Authors,

Thank you for submitting the revised version of your manuscript PGPH-D-25-02034R1. After carefully reviewing the updated document alongside the original reviewer report, I confirm that you have thoroughly and effectively addressed all comments raised during the previous review round.

Your revision demonstrates clear improvements in scientific clarity, methodological transparency, and presentation quality. In particular, the resolution of the sample-size inconsistency, the corrections to Figure 1 copyright requirements, the clarification of Table 2, the addition of the MCA figure, and the strengthened interpretation of sex- and age-related findings significantly enhance the robustness and readability of the manuscript.

The abstract, tables, and discussion are now more coherent and informative, and the manuscript aligns with journal standards.

I thank you for your careful and constructive revision. The manuscript is now suitable for publication pending minor editorial adjustments.

Sincerely,

**Do you want your identity to be public for this peer review?** For information about this choice, including consent withdrawal, please see our Privacy Policy

Reviewer #1: No
